# Investigation of Glypican-4 and -6 by Infrared Spectral Imaging during the Hair Growth Cycle

**DOI:** 10.3390/ijms24054291

**Published:** 2023-02-21

**Authors:** Charlie Colin-Pierre, Valérie Untereiner, Ganesh D. Sockalingum, Laurent Ramont, Stéphane Brézillon

**Affiliations:** 1Laboratoire de Biochimie Médicale et Biologie Moléculaire, Université de Reims Champagne-Ardenne, 51097 Reims, France; 2Matrice Extracellulaire et Dynamique Cellulaire-MEDyC, CNRS UMR 7369, 51097 Reims, France; 3BASF Beauty Care Solutions France SAS, 54425 Pulnoy, France; 4PICT, Université de Reims Champagne-Ardenne, 51097 Reims, France; 5Université de Reims Champagne-Ardenne, BioSpecT EA7506, UFR de Pharmacie, 51097 Reims, France; 6Service Biochimie-Pharmacologie-Toxicologie, CHU de Reims, 51097 Reims, France

**Keywords:** glycosaminoglycans, glypicans, hair follicle growth, infrared imaging

## Abstract

The expression of glypicans in different hair follicle (HF) compartments is still poorly understood. Heparan sulfate proteoglycans (HSPGs) distribution in HF is classically investigated by conventional histology, biochemical analysis, and immunohistochemistry. Our previous study proposed a novel approach to assess hair histology and glypican-1 (GPC1) distribution changes in the HF at different phases of the hair growth cycle using infrared spectral imaging (IRSI). We show in the present manuscript for the first time complementary data on the distribution of glypican-4 (GPC4) and glypican-6 (GPC6) in HF at different phases of the hair growth cycle using IR imaging. Findings were supported by Western blot assays focusing on the GPC4 and GPC6 expression in HFs. Like all proteoglycan features, the glypicans are characterized by a core protein to which sulfated and/or unsulfated glycosaminoglycan (GAG) chains are covalently linked. Our study demonstrates the capacity of IRSI to identify the different HF tissue structures and to highlight protein, proteoglycan (PG), GAG, and sulfated GAG distribution in these structures. The comparison between anagen, catagen, and telogen phases shows the qualitative and/or quantitative evolution of GAGs, as supported by Western blot. Thus, in one analysis, IRSI can simultaneously reveal the location of proteins, PGs, GAGs and sulfated GAGs in HFs in a chemical and label-free manner. From a dermatological point of view, IRSI may constitute a promising technique to study alopecia.

## 1. Introduction

The link between heparan sulfate proteoglycan (HSPG) and physiopathology regulation of hair follicle (HF) has been suggested by different studies. Indeed, several reports have highlighted the evolution of the distribution of HSPGs according to the hair growth cycle phases [1,2,3,4]. Others have shown the benefits of the treatment based on proteoglycans (PG) on the hair shaft growth [5,6,7,8]. Many cells constituting the HF (keratinocytes of outer root sheath (ORS), fibroblasts of dermal papilla, transit-amplifying (TA) cells of the matrix and hair stem cells) express glypicans (GPCs). GPCs are membrane glycosylphosphatidylinositol (GPI) anchored HSPGs. They are able to sequester many growth factors by their GAG chains or to interact with different molecules with their core protein [9,10,11,12,13,14]. To our knowledge, no study had been conducted on GPC distribution in HF in different phases (anagen, catagen and telogen) of the hair growth cycle before our previous paper [3]. The latter focused on glypican-1 (GPC1) distribution, which is highly expressed in HF, and its GAG chains [10]. Hair shaft growth exhibits similar mechanisms to embryogenesis. Moreover, GPC4 and 6 are expressed in different organs during embryogenesis such as kidney/brain and ovary, respectively [15]. Thus, in the present paper, we show complementary data on the distribution of glypican-4 (GPC4) and glypican-6 (GPC6) in HF at different phases of hair growth cycle.

Our previous results have demonstrated the ability of infrared (IR) spectral imaging to identify the different tissue structures of HF and to show the protein, PG, GAG and sulfated GAG distribution in these structures. In order to take our investigation further, we have investigated the distribution of GPC4 and GPC6 in HF at different cycle phases using IR imaging. In addition, the GPCs expression has been studied by Western blot.

## 2. Results and Discussion

### 2.1. IR Correlation Maps Highlight the Spatial Distribution of GPC4 and GPC6 at Different Phases of the Hair Growth Cycle

Figure 1A shows the visible image of a hair follicle with the different histological structures. In order to characterize the GPC4 and 6 in the hair follicles at different phases of the cycle, IR correlation maps were calculated using their representative mean spectra.

GPC4 was mainly detected in the germinative area of the hair matrix and in inner root sheath (IRS) for all phases of the cycle although GPC4 correlation seemed more important in the telogen phase in these structures (see arrowheads in Figure 1B–D). GPC4 was also detected in the ORS of the three phases with a different distribution. Indeed, in the telogen phase, the distribution appeared more continuous along the ORS while it was more punctuated in anagen and catagen phases. Concerning GPC6, it was mainly detected in the germinative zone of the hair matrix, in IRS and ORS for all phases of the hair cycle although GPC6 correlation seemed more important in the telogen phase in these structures (see arrowheads in Figure 1B–D). GPC6 was also detected in the differentiation zone of the hair matrix in catagen and telogen phases with a greater correlation in the telogen phase.

These results demonstrate that the telogen phase is characterized by an increased correlation of GPC4 and GPC6. However, in contrast to GPC4, the correlation maps suggest a progressive increase of GPC6 expression from the anagen phase to the telogen phase. It is important to note that GPC4 is mainly detected in germinative zone of the matrix and also in a punctuated form in the ORS while its correlation in hair shaft is very weak or undetectable. Thus, GPC4 could be an indicator for monitoring the differentiation of stem cells. Indeed, progenitor cells are produced from hair stem cells located in the ORS [16,17,18,19,20]. These progenitor cells migrate in the ORS to the germinative zone of the matrix surrounding the dermal papilla where they differentiate into TA cells of the matrix. These in turn differentiate into cells of the IRS and the hair shaft. GPC4 distribution previously observed in the different structures of the hair follicle seems to correspond to the location of these different cell populations. The formation of these different cell populations involves different mechanisms such as self-renewal, proliferation, migration, and differentiation. It is interesting to note that GPC4 is known to be involved in these processes in the case of the regulation of vertebrate embryonic and adult stem cell behavior occurring during embryogenesis, regeneration and homeostasis of tissues and organs. For example, GPC4 induces the migration of the lateral line collective cells essential for zebrafish embryogenesis by regulating Wnt/β-catenin and FGF signaling [21]. In mouse neuroepithelial cells, GPC4 inhibits differentiation and promotes their self-renewal by regulating Wnt/β-catenin signaling [22] and sequestrating FGF2 [23]. In contrast, in the case of Xenopus neurulation, GPC4 interacts with FGF2 to promote its binding to its receptor resulting in the regulation of forebrain patterning [24]. With regards to GPC6, its strong correlation in catagen and telogen phases in the matrix differentiation zone could be associated with the presence of the secondary hair germ formed during the catagen phase and still present in the telogen phase [25]. This result suggests a role of GPC6 in the formation of the secondary hair germ and thus in hair follicle stem cell differentiation. Comparatively, in the case of long bone growth, GPC6 promotes the interaction between Hedgehog and its receptor leading to chondrocytes differentiation [26]. 

### 2.2. Differentiation of Different Phases of the Hair Growth Cycle by GPC4 and GPC6 Expression

After analysis of GPC4 and GPC6 distribution in the HFs at different phases of the hair cycle, the protein expression of these two GPCs was also analyzed on hair follicles in the anagen, catagen and telogen phases (Figure 2). As shown in Figure 2A, the structures of the two molecules are very similar with 63% of homology of the amino acid of the core proteins [15]. Both exhibit a GPI anchor and HS GAG chains.

Protein analysis showed that cleaved, anchored and glycanated forms of GPC4 and GPC6 exhibited different expression patterns with respect to the different phases of the hair cycle. Indeed, the two cleaved forms of GPC4 were more abundant in the HF in the catagen phase compared to the anagen phase (average increase of 30%) and even more abundant in the telogen phase (average increase of 48%). Interestingly, the glycanated form and the 57 kDa anchored isoform of GPC4 were detected only in the HF in the catagen and telogen phases but with a higher expression in the telogen phase (increase of 94% for the glycanated form and of 60% for the 57 kDa anchored isoform compared to the anagen phase) (Figure 2B). Similarly, the cleaved, anchored and glycanated forms of GPC6 were more abundant in HFs in the telogen phase than in the catagen phase (increase of 58%, 60% and 10%, respectively). However, the different forms of GPC6 were not detected in the anagen phase (Figure 2C).

These results are in agreement with those obtained by IR imaging. Indeed, the highest expression of the different forms of GPC4 and GPC6 in the telogen phase confirms the strong level of correlation of GPC4 and GPC6 observed in the HF sections in this phase. In contrast to the IR spectral analysis, the protein assay by Western blot shows a very low or barely detectable expression of GPC4 and 6 in anagen HFs. Indeed, this experiment presents a lower detection level compared to the IR analysis but provides the ability to analyze the different forms of GPC. Both experiments are complementary and demonstrate the evolution of GPC4 and 6 expression and distribution during the hair growth cycle, thus providing strong evidence on the role of these GPCs in hair follicle physiology.

## 3. Materials and Methods

Figure 3 describes the experimental steps of this study.

### 3.1. Preparation of Hair Follicle for Infrared Spectral Acquisition

The HF were extracted and prepared for infrared imaging as described in our previous paper [3]. Briefly, each HF was embedded in cryoprotective Tissue-Tek^®^ O.C.T ^TM^ (Sakura, Alphen aan den Rijn, The Netherlands) and frozen at −80 °C. The longitudinal sections of HFs were performed at 7 µm thickness using a cryostat (Leica Biosystems, Nanterre, France) and deposited onto IR transparent calcium fluoride (CaF_2_) substrates (Crystran Ltd., Dorset, UK) for imaging. For all our experiments, four donors were involved with two to three hair samples on average per donor. For each hair sample, two to three sections were obtained. More precisely, 32 anagen, 28 catagen, 31 telogen HF sections were analyzed.

Spectral images of whole HFs sections were acquired using the Spotlight 400 (PerkinElmer, Villebon-sur-Yvette, France) IR imaging device in transmission mode, with a pixel resolution of 6.25 µm × 6.25 µm, a spectral resolution of 4 cm^−1^, and using 16 scans per pixel on the spectral range 4000–900 cm^−1^. For each section, a spectrum in an area of CaF_2_ substrate without sample or OCT compound was acquired at 90 scans/pixel and was automatically removed from each pixel of the image. For each image, the atmospheric contribution was corrected with the PerkinElmer software (Spectrum Image 1.7.1).

Recombinant human GPC4 and GPC6 (R&D Systems, Minneapolis, MN, USA), containing both the core protein and the GAG chains, were reconstituted at 100 μg/mL in sterile PBS, and a drop of 2 µL was deposited on a CaF_2_ substrate, dried before spectral image acquisition using the same parameters as above. All spectra of each image were averaged to obtain a representative mean spectrum of GPC4 and GPC6 standards.

### 3.2. Correlation of Infrared Images of HFs Using Representative Mean Spectrum of Glypican-4 and Glypican-6

The mean spectra of GPC4 and GPC6 were correlated pixel by pixel with the HF images using the Spectrum Image 1.7.1 software. The correlation maps obtained indicate the distribution of these molecules with a color scale ranging from 0 (dark color) to 1 (white color) corresponding to low and high correlation levels, respectively. In this analysis, the correlation was performed with respect to the IR spectral window 1800–900 cm^−1^ covering the protein and/or polysaccharide regions.

### 3.3. Western Blot

Total proteins of HFs were extracted from a pool of six HFs and were deposited onto polyacrylamide gels as previously described [3,27]. The primary antibodies GPC4, 13048-1-AP (Proteintech, Rosemont, IL, USA), and GPC6, AF2845 (R&D Systems, Minneapolis, MN, USA), were used. The appropriate peroxidase-coupled secondary antibodies (1/10,000) were the anti-rabbit NA934V (GE Healthcare Life Sciences, Marlborough, MA, USA) and the anti-goat A9452 (Sigma-Aldrich, Saint Louis, MO, USA).

## 4. Conclusions

The results obtained from IR spectral imaging provide chemical, morphological and quantitative information which were complemented by protein qualitative analyzes using Western blot. Thus, IR spectral imaging provides a non-invasive, chemical- and label-free method to analyze follicle physiopathology and the hair growth cycle. In a single analysis, it can simultaneously reveal the location of proteins, PGs, GAGs and sulfated GAGs in HFs. The analysis of GPC4 and GPC6 by these two techniques showed a different expression and distribution depending on the phases of the hair cycle. In particular, the accumulation and/or the appearance of the different forms of GPC4 and GPC6 during the three phases suggest the important role of these in the regulation of stem cells behavior. These GPCs could be potential markers of interest for monitoring hair stem cell differentiation.

Moreover, from a dermatological point of view, IRSI may constitute a promising technique to study alopecia.

## Figures and Tables

**Figure 1 ijms-24-04291-f001:**
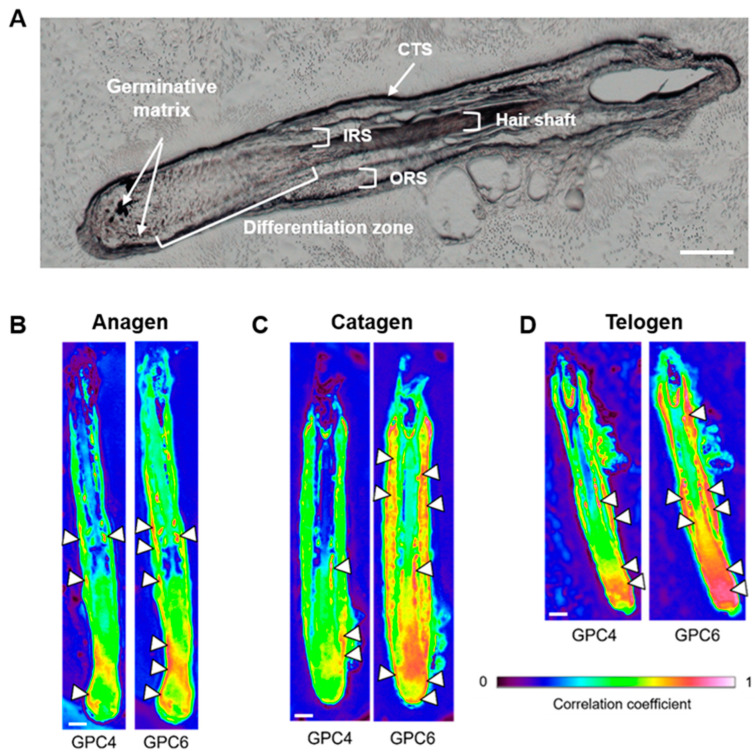
Glypican-4 (GPC4) and glypican-6 (GPC6) IR correlation maps in hair follicle. (**A**) White light image of hair follicle, (**B**–**D**) Correlation maps of hair follicles in the anagen (**B**), catagen (**C**) and telogen (**D**) phases using the mean IR spectra of human recombinant GPC4 and GPC6 (spectral range 1800–900 cm^−1^), respectively. A high level of correlation is indicated with arrowheads. (CTS) connective tissue sheath, (IRS) inner root sheath and (ORS) outer root sheath. Scale bar: 100 µm.

**Figure 2 ijms-24-04291-f002:**
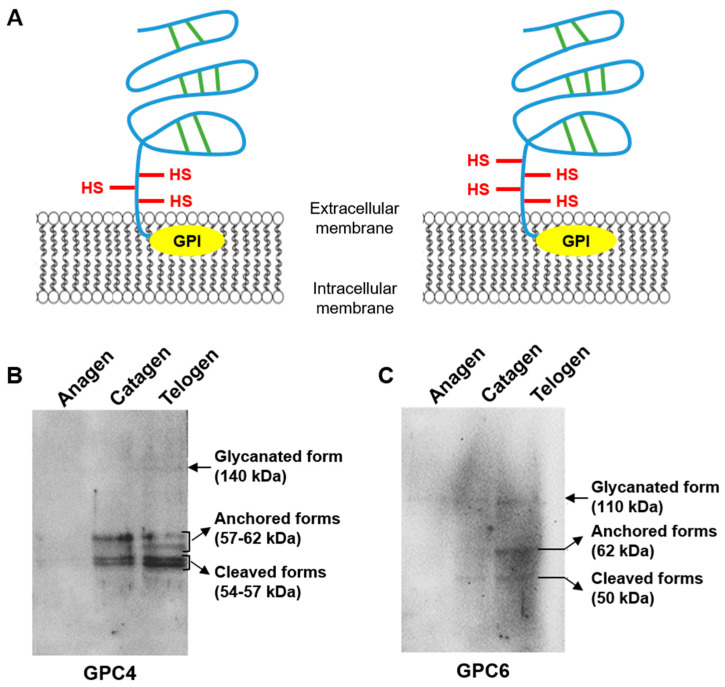
Schematic representation and expression of human glypican-4 (GPC4) and glypican-6 (GPC6) during hair growth cycle. (**A**) Schematic representation of GPC4 (left side) and GPC6 (right side). Expression of GPC4 (**B**) and GPC6 (**C**) in hair follicles (n = 6) during the anagen, catagen and telogen phases of the hair growth cycle analyzed by Western immunoblotting. Disulfide bond (green line), heparan sulfate (red line) and glycosylphosphatidylinositol (GPI) anchor (yellow ovoid).

**Figure 3 ijms-24-04291-f003:**
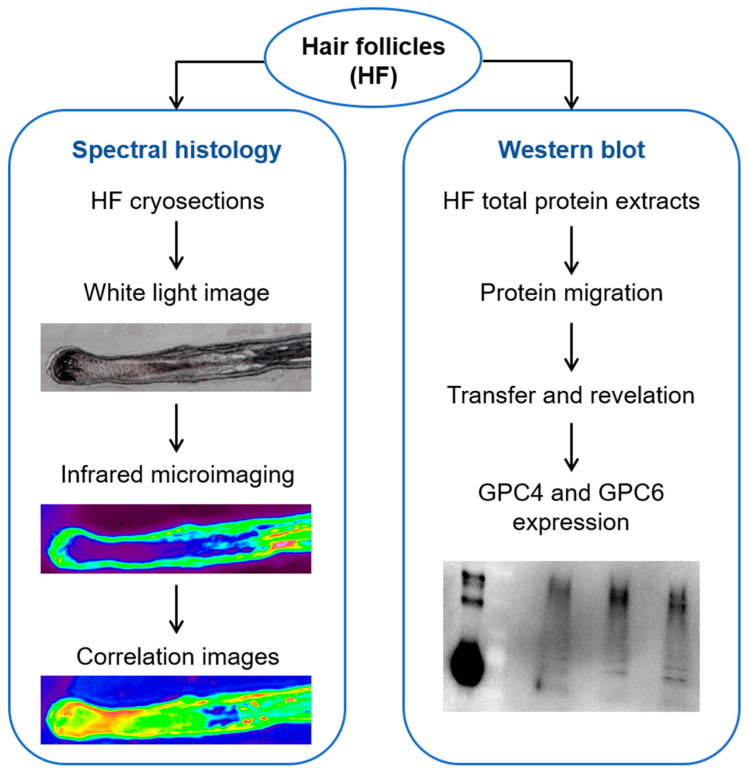
Workflow of hair follicle methodological approach. Pictures shown here are only illustrated examples.

## Data Availability

Not applicable.

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
