# Peer review of "Investigation of Glypican-4 and -6 by Infrared Spectral Imaging during the Hair Growth Cycle"

_ijms, 2023, doi:10.3390/ijms24054291_

Round 1

Reviewer 1 Report

The manuscript entitled "Investigation of Hair Histology and Proteoglycans Distribution 2 by Infrared Spectral Imaging Focus on Glypican-4 and -6 during the Hair Growth Cycle" submitted by Charlie Colin-Pierre et all is prepared as a communications. The manuscript present the results of combining two different method for studying the hair structure and its proteoglycans distributions. The methodology is mostly  based on Infrared spectroscopic imaging IRIS and western blot analysis.

The manuscript is clear and well prepared anyway some comments are below:

1)      The abbreviation of IRSI is infrared spectral imaging is not so popular as FTIR imaging. Even authors in the methodology part described that experiments were measured using Spotlight 400 (Perki-130 nElmer, Villebon-sur-Yvette, France).  

Also it's no clear what IRIS means when the abbreviation is used for the first time – abstract line line 25.  It should be explained.

2)      Figure 1 part A – the scale bar describing the  image size of visible image is missing, please add this  to improve quality of the manuscript

Reviewer 2 Report

It is important to represent the structural features of the studied molecules. It is also necessary to improve the quality of the Western blot and/or to use PCR-analysis.

Round 2

Reviewer 2 Report

It would be interesting to use additional method such as PCR and to perform statistical correlative analysis.